# MLP-BASED ARCHITECTURE WITH VARIABLE LENGTH INPUT FOR AUTOMATIC SPEECH RECOGNITION

## ABSTRACT

We propose multi-layer perceptron (MLP)-based architectures suitable for variable length input. Recently, several such architectures that do not rely on self-attention have been proposed for image classification. They achieve performance competitive with that of transformer-based architectures, albeit with a simpler structure and low computational cost. They split an image into patches and mix information by applying MLPs within and across patches alternately. Due to the use of MLPs, such a model can only be used for inputs of a fixed, pre-defined size. However, many types of data are naturally variable in length, for example, acoustic signals. We propose three approaches to extend MLP-based architectures for use with sequences of arbitrary length. In all of them, we start by splitting the signal into contiguous tokens of fixed size (equivalent to patches in images). Naturally, the number of tokens is variable. The two first approaches use a gating mechanism that mixes local information across tokens in a shift-invariant and length-agnostic way. One uses a depthwise convolution to derive the gate values, while the other relies on shifting tokens. The final approach explores non-gated mixing using a circular convolution applied in the Fourier domain. We evaluate the proposed architectures on an automatic speech recognition task with the Librispeech and Tedlium2 corpora. Compared to Transformer, our proposed architecture reduces the WER by $1.2/0.3\,\%$ on Librispeech test-clean/test-other set, and $1.6/1.6\,\%$ on Tedlium2 dev/test set, using only $86.4\,\%$ of the parameters. In addition, a hybrid of our proposed architecture and self-attention module reduces the WER by $1.9/3.4\,\%$ on Librispeech test-clean/test-other set, and $1.8/1.6\,\%$ on Tedlium2 dev/test set, using only $75.3\,\%$ of the parameters.

## 1   INTRODUCTION

Self-attention, the well-known building block of Transformer (Vaswani et al., 2017), appeared in natural language processing (NLP) where it caused a breakthrough. Soon enough, it propagated to the fields of computer vision and automatic speech recognition (ASR). In particular, recent end-to-end ASR systems based on self-attention architecture, e.g., Transformer (Karita et al., 2019a) and Conformer (Gulati et al., 2020), provide state-of-the-art performance.

Recently, several architectures entirely based on MLP have been proposed in the area of computer vision. MLP-based architectures have a simple structure, and achieve performance competitive with that of Transformer-based architectures, despite having fewer parameters and lower computational complexity. They split an image into patches and reshape it into a (channels × patches) matrix used as input. An illustration of the process, and its analog for variable length sequences, is shown in Figure 1. MLP-based architecture such as MLP-Mixer (Tolstikhin et al., 2021) and gMLP (Liu et al., 2021) consist of MLP across the channel dimension and MLP across the patch dimension (also referred to as spatial dimension in computer vision tasks). MLP across the channel dimension mixes information between channels like a typical feed-forward network (FFN). MLP across the patches dimension mixes information between patches. All these different works demonstrate that this process capture sufficient information and that self-attention is not always necessary. The small model size and low computational cost of the MLP-based architecture are also useful for ASR. For acoustic data, the input sequence is typically first split into contiguous or overlapping blocks and transformed to some kind of frequency domain representation, e.g. a Mel-spectrogram. Then, the time and frequency dimensions are analogous to patch and channel, respectively, for images. We

Figure 1: Illustration of the difference between patching for fixed-size images (left) and variable length sequences (right). We rename patches as *tokens* to adapt to the semantics of sequences.

adopt the terminology of "token" to describe the vector of channel values at a given time. It is more apt for sequences and consistent with that used in the MLP-Mixer and gMLP works. Now, in contrast to images, sequences will produce inputs with a variable number of tokens, making it impractical to apply an MLP directly to this dimension.

In this paper, we propose three approaches that can work with variable length inputs. First, the input sequences are broken down into contiguous chunks that we call tokens, as explained earlier and shown in Figure 1. Building on previous approaches, we propose three new token mixing units. Two of them rely on gating, where the input is split into two parts, one is transformed and then multiplied with the other. In the first kind of unit, Convolutional MLP (C-MLP), we use a depthwise convolution to transform the input. This operation is locally MLP-like along the token dimension and makes the network shift invariant. The second approach, Temporal-Shift MLP (TS-MLP), concatenates shifted parts of the input. The third approach is the non-gating one. In this case, we apply a simple depthwise convolution, thus mixing tokens with their local neighbors. However, we make use of the fast Fourier transform (FFT) to do so efficiently. This approach is termed Fourier MLP (F-MLP). We applied these MLP-based methods to the connectionist temporal classification (CTC) (Graves et al., 2006) ASR model and evaluated them on two datasets, Librispeech (Panayotov et al., 2015) and Tedlium2 (Rousseau et al., 2014). We found different trade-offs between accuracy in terms of word error rate (WER), the number of parameters, and inference speed. However, when matching the number of parameters, the proposed MLP architectures consistently outperform self-attention based models, decreasing the WER by as much as $3\%$.

**Related works.**

Recently, end-to-end (E2E) ASR models, where a single neural network produces a transcription directly from the input acoustic features, have come to dominate benchmarks. The E2E ASR model consists of an encoder that converts audio input into a latent representation and a decoder that converts the latent representation into text output (token). To build the encoder-decoder architecture, attention based encoder-decoder models such as Listen, Attend and Spell (LAS) (Chan et al., 2015) and recurrent neural network transducer(RNN-T) (Graves, 2012) have shown high performance, but they suffer from slow inference speed due to the way they auto-regressively output tokens. Another choice to build E2E ASR is non-autoregressive models with the audio encoder and a decoder using CTC (Graves et al., 2006). In recent years, many studies have been conducted on CTC-based models, and reported results demonstrate high inference speed for a performance comparable to that of attention-based encoder-decoder models. For the architecture of the audio encoder, recurrent neural networks (RNNs) and Convolution neural networks (CNNs) have been employed (Graves & Jaitly, 2014; Li et al., 2019; Kriman et al., 2020). More recently, models applying the self-attention based architecture (Karita et al., 2019a; Zhang et al., 2020; Gulati et al., 2020) have shown promising performance and become the de facto architecture. However, self-attention is expensive in terms of memory and computation, and reduction of both is an important endeavor.

Next, we summarize the recently proposed MLP-based architectures. MLP-Mixer (Tolstikhin et al., 2021) consists of channel-mixing MLPs and token-mixing MLPs. It alternately applies these MLPs and mixes information between channels and information between tokens. gMLP (Liu et al., 2021) also consists of MLPs across the channel and token dimensions. To more effectively mix tokens, it proposes a spatial gating unit (SGU), a module containing MLP across the token dimension and used to modulate the output of the unit. $S^2$-MLP (Yu et al., 2021a) and $S^2$-MLPv2 (Yu et al., 2021b) mix information across the token dimension by shift operation along the width dimension and height dimension called spatial-shift operation. GFNet (Rao et al., 2021) efficiently applies learnable filters

in the frequency domain. It performs a 2D Fourier transform of the input image, multiplies by the filters, and transforms back to the image domain. Then, MLP is applied across the channels. In the area of NLP, FNet (Lee-Thorp et al., 2021) has been proposed. It applies a 2D Fourier transform to the input data, retains the real part, and follows with MLP across channels.

**Contributions.** Our contributions in this work are as follows. (i) We propose three MLP-based architectures suitable for sequences of arbitrary lengths. (ii) We evaluate our architecture for ASR on two different datasets. We show that the proposed C-MLP architecture outperforms the celebrated self-attention, both in accuracy and inference speed. To the best of our knowledge, this is the first time that MLP-based architecture is applied to ASR.

## 2 CONVENTIONAL MLP-BASED ARCHITECTURES

The so-called MLP-based architectures are built using two modules, each based on simple MLPs: the channel mixing and token mixing modules. The channel mixing module typically consists of linear projections on the channel dimension for each token, similar to the point-wise feed-forward network in the Transformer. The token mixing module mixes the information in the token dimension as a replacement for self-attention, and how to design it is one of the keys to the MLP-based models. We explain conventional architectures dividing them into two types according to how they stack the modules: MLP-mixer and gMLP types, corresponding to the architectures proposed in (Tolstikhin et al., 2021) and (Liu et al., 2021), respectively. We start by describing the outer construction, i.e., how the channel and token mixing modules are combined. Then, we focus on the inners of a few kinds of token mixing module. Throughout, we omit layer normalization, residual connections, and bias of linear projections for the sake of brevity. Figure 2 shows diagrams of all the parts.

### 2.1 OUTER CONSTRUCTION

**MLP-Mixer type.** Figure 2 (a) shows the MLP-Mixer type which first applies the token-mixing module and then applies the channel-mixing module. It is adopted by its namesake (Tolstikhin et al., 2021), and GFNet (Rao et al., 2021). Let $\mathbf{X_{in}} \in \mathbb{R}^{D \times N}$ denote the input data matrix with $D$ channels and $N$ tokens. It first goes through the token mixing module,

$$\mathbf{X} = \text{TokenMixingModule}(\mathbf{X_{in}}) \in \mathbb{R}^{D \times N}, \tag{1}$$

where $\text{TokenMixingModule}()$ is an MLP-based module that usually consists of multiple linear projections on the token dimension. How to design this module is one of the key considerations for MLP-based models. Conventional approaches are introduced in Section 2.2.

Next, the output of the token-mixing module, $\mathbf{X}$, is fed to the channel-mixing module. The channel-mixing module is the same as the point-wise feed forward network in Transformer, which performs two linear projection across the channel dimension,

$$\mathbf{X_{out}} = \mathbf{W_2}\sigma(\mathbf{W_1 X}) \in \mathbb{R}^{D \times N}, \tag{2}$$

where $\mathbf{W_1} \in \mathbb{R}^{D' \times D}$ and $\mathbf{W_1} \in \mathbb{R}^{D \times D'}$ are the weights of the linear projections. We remark that the construction of the MLP-Mixer type is similar to that of a Transformer, with the token-mixing replacing self-attention. This highlights the critical importance of token mixing.

**gMLP type.** Figure 2 (b) shows the gMLP type where the two layers of the channel-mixing module are split and the token-mixing module placed in-between. This is the structure adopted by gMLP, $S^2$-MLP and $S^2$-MLPv2. Let $\mathbf{X_{in}} \in \mathbb{R}^{D \times N}$ denote the input data with $D$ channels and $N$ tokens. First, linear projection on channel dimension $D$ to $D'$ is applied to $\mathbf{X_{in}}$,

$$\mathbf{X} = \sigma(\mathbf{W_1 X_{in}}) \in \mathbb{R}^{D' \times N}, \tag{3}$$

where $\mathbf{W_1} \in \mathbb{R}^{D' \times D}$ refers to weights of linear projection. Next, the token-mixing module follows,

$$\mathbf{X'} = \text{TokenMixingModule}(\mathbf{X}) \in \mathbb{R}^{D'' \times N}. \tag{4}$$

Finally, linear projection across the channel dimension $D''$ to $D$ is applied as

$$\mathbf{X_{out}} = \mathbf{W_3 X'} \in \mathbb{R}^{D \times N}, \tag{5}$$

where $\mathbf{W_3} \in \mathbb{R}^{D \times D''}$ is the weight matrix of the linear projection.

Here, additional linear projections can be utilized for mixing the channel information. For example, $S^2$-MLP stacks two additional linear projections after the token-mixing module.

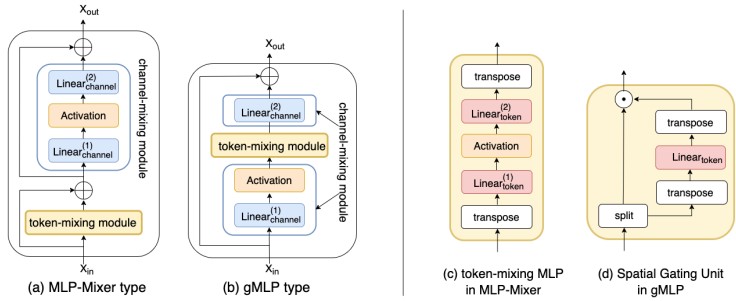

Figure 2: Overview of (a) MLP-Mixer type, (b) gMLP type, (c) token-mixing MLP in MLP-Mixer, and (d) Spatial Gating Unit in gMLP.

## 2.2 CONVENTIONAL TOKEN-MIXING MODULES AND CHALLENGES FOR ASR

There have been several studies on how to design the token-mixing module, such as token-mixing MLP in MLP-mixer, spatial gating unit (SGU) in gMLP, Fourier Domain Mixing in GFNet (Rao et al., 2021) and FNet (Lee-Thorp et al., 2021). In this paper, we introduce these modules and discuss the issues in applying them to variable sequence length input data. In the following, the input is always assumed to be a data matrix with $D$ channels and $N$ tokens, $\mathbf{X} \in \mathbb{R}^{D \times N}$.

**Token-mixing MLP.** The token-mixing MLP first transposes the input $\mathbf{X}$ and performs linear projection across the token dimension to mix the token information,

$$\mathbf{X}' = (\mathbf{W_2}\sigma(\mathbf{W_1}\mathbf{X}^\mathsf{T}))^\mathsf{T} \in \mathbb{R}^{D \times N}, \tag{6}$$

where $\mathbf{W_1} \in \mathbb{R}^{N' \times N}$ and $\mathbf{W_2} \in \mathbb{R}^{N \times N'}$ refer to weights of linear projections, and $\sigma(\cdot)$ is an element-wise activation function. Albeit the components are the same as the channel-mixing module of Equation 2, the linear projections are done on the token dimension by transposing the input. This module is shown in Figure 2 (c).

**Spatial Gating Unit.** The structure of the SGU is illustrated in Figure 2 (d). The SGU first splits the input $\mathbf{X}$ into $\mathbf{X}_\mathrm{r} \in \mathbb{R}^{\frac{D}{2} \times N}$ and $\mathbf{X}_\mathrm{g} \in \mathbb{R}^{\frac{D}{2} \times N}$. In gMLP, $\mathbf{X}_\mathrm{r}$ is set as the first half of $\mathbf{X}$ and $\mathbf{X}_\mathrm{g}$ for the second half of $\mathbf{X}$. Then, it performs a gating operation. The output of SGU $\mathbf{X}'$ is,

$$\mathbf{X}' = \mathbf{X}_\mathrm{r} \odot \mathbf{H}_\mathrm{SGU} \in \mathbb{R}^{\frac{D}{2} \times N}, \quad \text{with} \quad \mathbf{H}_\mathrm{SGU} = (\mathbf{W_2}\mathbf{X}_\mathrm{g}^\mathsf{T})^\mathsf{T} \in \mathbb{R}^{\frac{D}{2} \times N}, \tag{7}$$

where $\odot$ denotes element-wise multiplication, and $\mathbf{W_2} \in \mathbb{R}^{N \times N}$ refers to the weights of the linear projection. This gating operation mixes the information of the token effectively.

The effectiveness of the SGU has been experimentally shown with image and language experiments compared to non-gating linear projection and other variants.

**Fourier Domain Mixing.** GFNet and FNet use the discrete Fourier transform (DFT) in the token-mixing module. GFNet applies a fixed length learnable filter by direct multiplication in the frequency domain. FNet mixes tokens by the application of a single forward 2D DFT, retaining only the real part of the output.

## 2.3 CHALLENGES FOR ASR.

The sizes of the weight matrices in Equation 6 and 7 are fixed and have to match the number of tokens in the input. Similarly, the learnable filter in GFNet is of fixed size equal to the length of the input sequence. These approaches are thus not suitable for variable length input. Nevertheless, given a fixed dataset, they could be applied by zero-padding all sequences to the size of the longest one. However, there are a number of shortcomings. First and foremost, longer sequences that might be encountered at test time cannot be accommodated. Then, MLPs will introduce a large number of parameters. Moreover, they are not shift invariant, an important property for sequences. While GFNet is shift invariant and only requires a reasonable number of parameters, it still cannot deal with sequences longer than the maximum size set. FNet has neither of the above limitations, however, we find its performance somewhat lacking in the experiments of Section 4.

Figure 3: Overview of (a) Convolutional Gating Unit, (b) Convolutional Gating Unit′, (c) Temporal-Shift Gating Unit, and (d) Temporal-Shift operation.

# 3  MLP-BASED ARCHITECTURE FOR VARIABLE LENGTH INPUT

We describe here our three proposed MLP-based architectures for ASR. These approaches are suitable for sequences of any length and shift invariant. As described in Section 2.2, the design of the token-mixing module is important to apply MLP-based architecture to ASR. We first propose three token-mixing modules for ASR in 3.1. Then, we describe the overall architecture in 3.2.

## 3.1  TOKEN-MIXING MODULE FOR VARIABLE LENGTH INPUT

### 3.1.1  CONVOLUTIONAL GATING UNIT (CGU)

**Convolutional Gating Unit (CGU).** The conventional token-mixing modules described in Section 2.2 mix information globally over the full extent of the data. However, for long sequences, we think it is sufficient to mix temporally local information. We do so by using a convolutional layer along the time dimension. We call this token-mixing module Convolutional Gating Unit (CGU). The structure of a CGU is shown in Figure 3 (a). CGU replaces linear projection across the token dimension in SGU (see Equation 7) with a depthwise convolution. Its output $\mathbf{H}_{\mathrm{CGU}}$ is,

$$\mathbf{H}_{\mathrm{CGU}} = \mathbf{K} \star \mathbf{X}_{\mathrm{g}} \in \mathbb{R}^{\frac{D}{2} \times N}, \tag{8}$$

where $\mathbf{X}_{\mathrm{g}} \in \mathbb{R}^{\frac{D}{2} \times N}$, $\mathbf{K} \in \mathbb{R}^{\frac{D}{2} \times k}$ is the $\frac{D}{2}$-dimensional kernel with kernel size $k$. The depth wise convolution operation is denoted by $\star$ and defined as,

$$(\mathbf{K} \star \mathbf{X})_{:,i} = \sum_{j=1}^{k} \mathbf{K}_{:,j} \odot \mathbf{X}_{:,k+i-j}. \tag{9}$$

**Convolutional Gating Unit′ (CGU′).** We also propose a variation where a linear projection across the channel dimension is applied to the filter $\mathbf{H}_{\mathrm{CGU}}$. In this case, information from both token and channel dimensions is mixed. The new filter $\mathbf{H}'_{\mathrm{CGU}}$ is formulated as follows,

$$\mathbf{H}'_{\mathrm{CGU}} = \mathbf{W}\mathbf{H}_{\mathrm{CGU}} \in \mathbb{R}^{\frac{D}{2} \times N}. \tag{10}$$

where $\mathbf{W} \in \mathbb{R}^{\frac{D}{2} \times \frac{D}{2}}$ is the weight matrix of the linear projection. This structure is illustrated in Figure 3 (b). We refer to this token-mixing module as Convolutional Gating Unit′ (CGU′)

### 3.1.2  TEMPORAL-SHIFT GATING UNIT (TSGU)

In CGU, we used learnable kernels for convolution across the token dimension. Next, we consider the case of performing a convolution using kernels with fixed parameters. If the input is a $D \times N$ matrix, then the $i$th row of the fixed kernel $\mathbf{K}_{\mathrm{shift}}$ is

$$\mathbf{k}_i = \begin{cases} [0, 0, 0, 0, 1] & \text{if } 1 \le i \le \frac{D}{2}, \\ [1, 0, 0, 0, 0] & \text{if } \frac{D}{2} < i \le D. \end{cases} \tag{11}$$

This corresponds to the shift operation proposed in S$^2$-MLP (Yu et al., 2021a). Half of the channels are shifted by two forward, and the other half by two backward along the time dimension. The time shift action of this kernel on the input signal is illustrated in Figure 3 (d). The size of the shift is a parameter but we fix it here to the value taken in the experiments. We propose the Temporal Shift

Gating Unit (TSGU) as a token-mixing module with a shift operation in the token direction. The inputs is split as in SGU, and the gating values derived as

$$\mathbf{H}_{\text{TSGU}} = \mathbf{K}_{\text{shift}} \star \mathbf{X}_g, \tag{12}$$

which replaces $\mathbf{H}_{\text{SGU}}$. This gating mechanism is parameter-free and mixes time information by applying progressive time shifts to the input signal.

### 3.1.3 FOURIER FILTER UNIT (FFU)

In both CGU and TSGU, we use a gating unit based on the SGU, as in gMLP. To investigate the usefulness of gating, we introduce a third architecture that does not use it. This is a circular depthwise convolution constructed as follows. It can be understood as similar to CGU, but without the split, and where all the input is processed through the depthwise convolution. The fixed size filters are defined in the time domain, but applied in the frequency domain. Thus, the filter is easily applied to sequences of any length by applying an FFT padded to the length of the input sequence to the filters, and working in the frequency domain. We call this token-mixing module Fourier Filter Unit (FFU). The output of the circular depthwise convolution is computed as follows,

$$\mathbf{Z} = \mathcal{F}^{-1} \left[ \mathcal{F}[\mathbf{X}] \odot \mathcal{F}[\mathbf{K}] \right] \in \mathbb{R}^{D \times N}, \tag{13}$$

where $\mathbf{X} \in \mathbb{R}^{D \times N}$ is the input signal and $\mathbf{K} \in \mathbb{R}^{D \times n}$ contains the filters in its rows. The FFT and its inverse are denoted $\mathcal{F}[\cdot]$ and $\mathcal{F}^{-1}[\cdot]$, respectively. We assume that the operation zero-pads $\mathbf{K}$ to size $D \times N$ to match $\mathbf{X}$.

GFNet (Rao et al., 2021) also applies element-wise multiplication to learnable filters and input features in the frequency domain. To apply GFNet to variable sequence length input, we have to find the maximum sequence length of the dataset and define filters to match that length. On the other hand, FFU defines filters in the time domain, zero-pads it to match the size of each input data, and perform FFT to the filter and input data. Therefore, FFU has fewer parameters than GFNet.

### 3.2 OVERALL ARCHITECTURE

We have proposed three types of token-mixing modules, but there is a choice in the outer construction, described in Section 2.1, to build the whole architecture. We have conducted preliminary experiments (See Appendix A) and finally propose the following four MLP-based architectures.

- **Convolutional MLP (C-MLP)**: CGU + gMLP type outer construction
- **Convolutional MLP′ (C-MLP′)**: CGU′ + gMLP type outer construction
- **Temporal-Shift MLP (TS-MLP)**: TSGU + gMLP type outer construction
- **Fourier MLP (F-MLP)**: FFU + MLP-Mixer type outer construction

## 4 EXPERIMENTS

### 4.1 EXPERIMENTAL SETUP

We applied our architectures to non-autoregressive CTC-based ASR. The experiments were conducted using the ESPNet tool kit (Watanabe et al., 2018). We use Transformer-encoder as self-attention based baseline, which is a strong baseline used in many prior works (Bai et al., 2020; Higuchi et al., 2021; Lee & Watanabe, 2021).

We also use FNet (Lee-Thorp et al., 2021) and GFNet (Rao et al., 2021) as MLP-based baseline. In order to justify the performance of the architecture itself, all parameters other than the architecture were kept the same in all experiments.

**Encoder layer structure.** For MLP-Mixer type architectures (FNet, GFNet, and F-MLP), input and output dimensions of the token-mixing module are set to 256, input dimension, hidden dimension, and output dimension of the channel-mixing module are set to 256, 1024, 256. For gMLP type architectures (C-MLP, C-MLP′, and TS-MLP), input and output dimensions of the first channel-mixing module are set to 256 and 1024, input and output dimensions of the token-mixing module

Table 1: Experimental results on Librispeech and Tedlium2.

| method | params | WER (Librispeech) | | WER (Tedlium2) | |
|---|---|---|---|---|---|
| | | test-clean | test-other | dev | test |
| *Baseline* | | | | | |
| Transformer-based model | 16.2M | 13.3% | 32.2% | 14.4% | 13.6% |
| FNet (Lee-Thorp et al., 2021) | 11.5M | 33.8% | 63.3% | 28.5% | 28.3% |
| GFNet (Rao et al., 2021) | 16.2M | 12.5% | 31.4% | 13.6% | 12.7% |
| *Ours* | | | | | |
| F-MLP | 11.5M | 16.0% | 37.0% | 16.3% | 15.8% |
| F-MLP+tiny attn | 13.8M | 12.1% | 30.8% | 13.2% | 13.0% |
| C-MLP | 9.3M | 13.4% | 33.8% | 14.5% | 13.5% |
| C-MLP+tiny attn | 12.2M | 11.4% | 28.8% | 12.6% | 12.0% |
| C-MLP$'$ | 14.0M | 12.1% | 31.9% | 12.8% | 12.0% |
| C-MLP$'$+tiny attn | 16.9M | **11.1%** | **28.3%** | **11.5%** | **11.4%** |
| TS-MLP | **9.1**M | 16.0% | 40.0% | 18.1% | 17.5% |
| TS-MLP+tiny attn | 12.1M | 13.5% | 32.6% | 14.1% | 13.5% |

are set to 512 and 256, input dimension, hidden dimension, and output dimension of the second channel-mixing module are set to 256, 1024, 256. For both types, we use Gaussian Error Linear Units (GELU) (Hendrycks & Gimpel, 2016) as activation function. Convolution kernel size of C-MLP and C-MLP$'$ and filter size of F-MLP are 15 and shift size of TS-MLP is 2. For Transformer encoder, we follow the architecture proposed in Karita et al. (2019b). The number of heads and input dimensions of the self-attention module are 4 and 256. The intermediate dimensions of the feed-forward network are 1024. For all models, we use two CNN-based subsampling layers before encoder layers. We set the subsampling rate to 0.25. All models consist of 18-layers.

**Tiny self-attention module.** gMLP (Liu et al., 2021) demonstrate that adding a small self-attention module to the SGU can improve performance at the cost of a modest increase in resource usage. The structure of an SGU with a tiny self-attention module is shown in Figure 4. The input of the tiny self-attention module is the input of the encoder. The output of the tiny self-attention module is added to the end of the right path of SGU. The tiny self-attention module has the same structure as the self-attention module in Transformer encoder, but its hidden dimension of linear projection $d$ and the number of

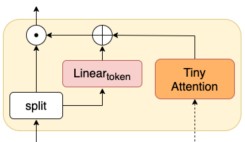

Figure 4: SGU with a tiny self-attention.

attention heads $n_{\text{head}}$ are small. We also experimented with the proposed token-mixing module combined with tiny attention. We set the tiny self-attention to $n_{\text{head}} = 1, d = 128$, while we set self-attention module in Transformer-based model to $n_{\text{head}} = 4, d = 256$.

**Datasets.** We measure performance on two datasets: Librispeech and Tedlium2. These two datasets are widely used for the evaluation of non-autoregressive ASR . Tedlium2 contains utterances from English Ted Talks, and we used the 207-hour training data. We used the standard validation and test sets for tuning hyper-parameters and evaluating performance, respectively. For Librispeech, we used the 100-hour subset for training set and standard validation and test sets as (Park et al., 2020; Zhao et al., 2020; Higuchi et al., 2021). Specifically, the validation and test sets of Librispeech are divided into "clean" and "other" based on the quality of the recorded utterances.

**Feature Extraction.** We use 80-dimensional log-mel-spectrogram features and 3-dimensional pitch features as inputs. For feature extraction, we use kaldi (Povey et al., 2011), and shift size and length of the window are set to $10\,\mathrm{ms}$ and $25\,\mathrm{ms}$, respectively. We apply SpecAugment (Park et al., 2019) and speed perturbations (Ko et al., 2015) for data augmentation. For Librispeech, we tokenize text into 300 subwords, and for Tedlium2, we tokenize text into 500 subwords. We created subwords with SentencePiece (Kudo & Richardson, 2018)

**Training and inference setup.** All models are trained for 50 epochs. We set the batch size to 64. The optimizer is Adam with $\beta_1 = 0.9$, $\beta_2 = 0.98$, $\epsilon = 10^{-9}$. The scheduling method for the learning rate is the same as (Vaswani et al., 2017), $\text{learning rate} = d^{-0.5} \cdot \min(\text{step\_num}, \text{step\_num} \cdot \text{warmup\_steps}^{-1.5})$, where we set warmup_steps and $d$ to 25000 and 1280, respectively. Dropout rate and label smoothing rate are set to 0.1. For inference, we use the model parameters obtained

Table 2: Comparison of model size and WER. In order to stabilize the learning of deep models, all models were trained using InterCTC (Lee & Watanabe, 2021), a regularization technique used in CTC-based ASR.

| method | layers | params | WER (Librispeech) | | WER (Tedlium2) | |
|---|---|---|---|---|---|---|
| | | | test-clean | test-other | dev | test |
| Transformer-based model | 9 | 9.1M | 14.1% | 32.9% | 15.2% | 14.6% |
| F-MLP | 15 | 9.9M | 15.5% | 37.3% | 17.3% | 16.4% |
| C-MLP | 18 | 9.3M | 11.5% | 31.2% | 13.6% | 12.7% |
| C-MLP$'$ | 12 | 10.0M | **11.4%** | 31.3% | 13.5% | **12.5%** |
| TS-MLP | 18 | 9.1M | 14.6% | 38.0% | 17.6% | 16.8% |
| F-MLP+tiny attn | 12 | 9.9M | 11.9% | 30.3% | 13.7% | 12.8% |
| C-MLP+tiny attn | 12 | 8.8M | 11.7% | 29.9% | 13.5% | 12.9% |
| C-MLP$'$+tiny attn | 9 | 9.4M | 11.6% | **29.3%** | **13.2%** | 12.8% |
| TS-MLP+tiny attn | 12 | 8.7M | 13.3% | 32.9% | 14.5% | 14.0% |
| Transformer-based model | 18 | 16.2M | 10.9% | 27.8% | 11.5% | 11.1% |
| F-MLP | 27 | 16.3M | 13.9% | 34.6% | 15.7% | 14.8% |
| C-MLP | 36 | 16.5M | 9.6% | 27.0% | 11.4% | **10.1%** |
| C-MLP$'$ | 21 | 16.0M | 10.1% | 28.5% | 11.4% | 10.5% |
| TS-MLP | 36 | 16.3M | 11.2% | 32.7% | 13.5% | 12.7% |
| F-MLP+tiny attn | 21 | 15.8M | 10.4% | 27.0% | 11.9% | 11.4% |
| C-MLP+tiny attn | 27 | 17.3M | 9.8% | 26.0% | 11.0% | 10.4% |
| C-MLP$'$+tiny attn | 18 | 16.9M | **9.4%** | **24.9%** | **10.5%** | 10.2% |
| TS-MLP+tiny attn | 27 | 16.3M | 9.9% | 26.2% | 11.0% | 10.3% |
| Transformer-based model | 36 | 30.4M | 9.9% | 25.8% | 10.7% | 9.7% |
| F-MLP | 51 | 29.0M | 13.1% | 32.5% | 16.6% | 15.4% |
| C-MLP | 72 | 31.1M | 8.6% | 25.0% | 9.8% | 8.8% |
| C-MLP$'$ | 42 | 30.0M | 9.1% | 26.0% | 9.5% | 8.7% |
| TS-MLP | 72 | 30.5M | 10.0% | 30.6% | 11.6% | 10.8% |
| F-MLP+tiny attn | 42 | 29.8M | 9.0% | 24.1% | 10.5% | 9.8% |
| C-MLP+tiny attn | 51 | 31.0M | **7.9%** | **21.0%** | **9.3%** | **8.5%** |
| C-MLP$'$+tiny attn | 33 | 29.4M | 8.1% | 21.8% | 10.2% | 9.3% |
| TS-MLP+tiny attn | 51 | 30.6M | 8.8% | 23.5% | 9.8% | 9.1% |

by averaging the 10 models with the best validation scores. The outputs are decoded by greedy decoding for CTC, without using any external language model.

## 4.2 RESULTS

**Main results.** Table 1 provides a comparison of the parameter sizes and word error rates (WER) on Librispeech and Tedlium2. In Table 1, we see that F-MLP is $71.0\%$ the size of GFNet, which also uses Fourier transform, and the Transformer-based model. Compared to the Transformer-based model, F-MLP degrades WER by $2.7/4.8\%$ on Librispeech test-clean/test-other sets and $1.9/2.2\%$ on Tedlium2 dev/test sets. However, F-MLP with a tiny self-attention improves WER by $3.9/6.2\%$ on Librispeech test-clean/test-other set and $3.1/2.8\%$ on Tedlium2 dev/test set from F-MLP and outperforms Transformer-based model and GFNet. In addition, even when combined with tiny self-attention, its model size is smaller than that of Transformer-based model and GFNet.

C-MLP achieves competitive performance with Transformer-based model with only $57.4\%$ of its parameters. C-MLP$'$ achieves the best WER in Table 1 and improves WER by $1.2/0.3\%$ on Librispeech test-clean/test-other set, and $1.6/1.6\%$ on Tedlium2 dev/test set. It increases the model size a little but is still only $86.4\%$ the size of the Transformer-based model. C-MLP with a tiny attention improves WER by $1.9/3.4\%$ on Librispeech test-clean/test-other set, and $1.8/1.6\%$ on Tedlium2 dev/test set, while using only $75.3\%$ of the parameters. TS-MLP, which can be said to be a special case of C-MLP, has the smallest number of parameters. TS-MLP has only $56.2\%$ of parameters of Transformer-based CTC while degrading WER by $2.7/7.8\%$ on Librispeech test-clean/test-other set and $3.7/3.9\%$ on Tedlium2 dev/test set.

**Model Scaling Analysis.** Table 2 shows the performance when the models are scaled to have approximately the same number of parameters. We evaluate three model sizes corresponding to the number of parameters for Transformer-based model with 9, 18, and 36 layers. Under most conditions, C-MLP shows the highest accuracy. When the number of layers is scaled up and deepened, C-MLP and TS-MLP show a large performance improvement compared to the other models. This may be because the number of layers of C-MLP and TS-MLP is larger than other models due to their smaller size. We conjecture that the larger number of layers allows better mixing between a wider range of locations, leading to improved representations.

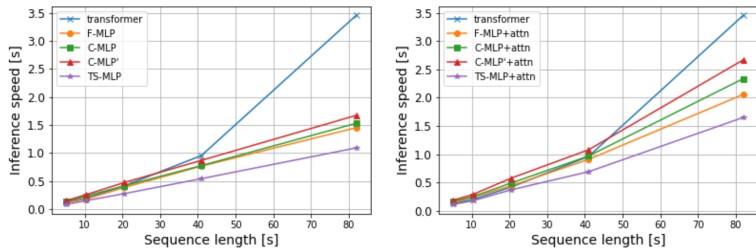

Figure 5: Comparison of inference speed for different sequence lengths

Table 3: Computational cost of architectures. $N$ is the sequence length and $D$ is the size of channel dimension of a feature. $l$ is the size of filters in Fourier Filter Unit. $k$ is the convolution kernel size.

| Method | computational complexity | number of parameters |
|---|---|---|
| linear projection | $ND^2$ | $D^2$ |
| self-attention | $4ND^2 + 2N^2D$ | $4D^2$ |
| Fourier Filter Unit | $ND\log_2 N + ND$ | $lD$ |
| Convolutional Gating Unit | $kND^2$ | $kD$ |
| Temporal Shift Gating Unit | $N + ND$ | - |
| tiny self-attention | $2ND^2 + 2N^2D$ | $2D^2$ |

**Computational Cost Analysis.** We investigate inference speed with randomly created 83-dimensional inputs with sequence lengths of 512, 1024, 2048, 4096, 8192 frames. A sequence of length 512 corresponds to $5.12\,\mathrm{s}$ of audio. We use an 18-layer encoder for every method. We conduct our experiments with a batch size 1 on a single Intel® Xeon® Gold 6230 CPU @ 2.10GHz with 4 cores. The reported times are the average of ten measurements. We show the inference speed for each input sequence length in Figure 5 and show computational characteristics of each architecture in Table 3. $N$ denotes the input sequence length and $D$ the number of channels. From Table 3, we see that when sequence length $N$ is small, the influence of the size of $D$ on operation complexity is strong. As shown in Figure 5, the inference speed of C-MLP and C-MLP′ is slower than Transformer-based model when $N$ is small. On the other hand, the influence of the size of $D$ on operation complexity is small as shown in Table 3. So, in Figure 5, we see that the inference speed of Transformer-based model is slower than any of our architectures when $N$ is large. TS-MLP, which can be said to be a special case of C-MLP, only performs a simple shift operation and element-wise multiplication. The TSGU has no parameters. Therefore, it achieves the fastest inference speed in Figure 5 regardless of the input sequence length. When combined with a tiny self-attention, the inference time increases, but the trend remains.

## 5 CONCLUSION

We proposed three new network architectures based on MLP-mixing for sequences of variable size. Each uses a different mechanism to mix information at long-range across the sequence. C-MLP relies on convolutional gating, TS-MLP on time-shift gating, and F-MLP simply on circular convolution without gating. Extensive experiments revealed that these MLP architectures are sufficient to outperform Transformer-based models for non-autoregressive ASR. Among the different models, C-MLP was the best, suggesting that gating is useful and necessary to squeeze out the best performance. When the proposed models alone were not sufficient, adding a tiny self-attention layer was enough to bridge the gap, while keeping the parameter count low. However, this was not always necessary when matching the number of parameters of the different networks. We thus conclude that all three proposed MLP-like architectures are not only suitable for ASR, but also highly practical due to their simplicity and good performance. In the future, we will explore their application to other tasks such as natural language processing or acoustic event detection.

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

## A  APPENDIX A: ARCHITECTURE TYPE EXPERIMENT

Our proposed token-mixing modules can be applied to either of two types of architectures described in 2.1: MLP-Mixer type and gMLP type. We conducted preliminary experiments to decide which type of architecture to adopt for each of the proposed token-mixing modules. We show the results of the preliminary experiments in Table 4.

Table 4: Comparison of architecture type and WER.

| architecture type | params | WER (Librispeech) | |
| | | test-clean | test-other |
| --- | --- | --- | --- |
| FFU + MLP-Mixer type (FMLP) | 11.5M | **16.0**% | **37.0**% |
| FFU + gMLP type | 9.2M | 18.1% | 41.1% |
| FFU + MLP-Mixer type (FMLP) + tiny attn | 13.8M | **12.1**% | **30.8**% |
| FFU + gMLP type | 12.2M | 13.2% | 32.1% |
| CGU + MLP-Mixer type | 11.5M | 13.6% | 34.6% |
| CGU + gMLP type (C-MLP) | 9.3M | **13.4**% | **33.8**% |
| CGU + MLP-Mixer type + tiny attn | 13.9M | 11.8% | 29.4% |
| CGU + gMLP type (C-MLP) + tiny attn | 12.2M | **11.4**% | **28.8**% |
| CGU′ + MLP-Mixer type | 12.7M | 13.6% | 34.4% |
| CGU′ + gMLP type (C-MLP′) | 14.0M | **12.1**% | **31.9**% |
| CGU′ + MLP-Mixer type + tiny attn | 15.1M | 11.9% | 29.9% |
| CGU′ + gMLP type (C-MLP′) + tiny attn | 16.9M | **11.1**% | **28.3**% |
| TSGU + MLP-Mixer type | 11.5M | 18.7% | 43.2% |
| TSGU + gMLP type (TS-MLP) | 9.1M | **16.0**% | **40.0**% |
| TSGU + MLP-Mixer type + tiny attn | 13.8M | 13.6% | 33.3% |
| TSGU + gMLP type (TS-MLP) + tiny attn | 12.1M | **13.5**% | **32.6**% |

## B  APPENDIX B: HYPERPARAMETERS FOR OUR EXPERIMENTS

We summarize the hyperparameters described in section 4 in Table 5 and Table 6.

Table 5: Hyperparameters

| hyperparameter | value |
| --- | --- |
| Epoch | 50 |
| batch size | 64 |
| dropout rate | 0.1 |
| label smoothing rate | 0.1 |
| subsampling rate | 0.25 |
| Optimizer | $Adam(\beta_1 = 0.9, \beta_1 = 0.98, \epsilon = 10^{-9})$ |
| learning rate | $d^{-0.5} \cdot \min(\text{step\_num}, \text{step\_num} \cdot \text{warmup\_steps}^{-1.5})$ |
| warmup steps | 25000 |
| $d$ | 1280 |
| window length (Feature Extraction) | 25ms |
| window shift (Feature Extraction) | 10ms |

Table 6: input and output channel dimensions of the architectures

|  | module | parameters |
|---|---|---|
| Transformer encoder | self-attention | $\text{num\_head} = 4, \text{hiddendim} = 256$ |
|  | Linear$^{(1)}$ in FFN | $\text{input} = 256, \text{output} = 1024$ |
|  | Linear$^{(2)}$ in FFN | $\text{input} = 1024, \text{output} = 256$ |
|  | Activation | GELU |
| MLP-Mixer type (F-MLP, GFNet, FNet) | token-mixing module | $input = 256, output = 256$ |
|  | Linear$^{(1)}_{\text{channel}}$ | $\text{input} = 256, \text{output} = 1024$ |
|  | Linear$^{(2)}_{\text{channel}}$ | $\text{input} = 1024, \text{output} = 256$ |
|  | Activation | GELU |
|  | filter size (F-MLP) | 15 |
|  | tiny self-attention | $\text{num\_head} = 1, \text{hiddendim} = 128$ |
|  | filter size (GFNet) | 512 |
| gMLP type (C-MLP, C-MLP′, TS-MLP) | Linear$^{(1)}_{\text{channel}}$ | $\text{input} = 256, \text{output} = 1024$ |
|  | token-mixing module | $\text{input} = 1024, \text{output} = 512$ |
|  | Linear$^{(2)}_{\text{channel}}$ | $\text{input} = 512, \text{output} = 256$ |
|  | Activation | GELU |
|  | convolution kernel size (C-MLP, C-MLP′) | 15 |
|  | shift size (TS-MLP) | 2 |
|  | tiny self-attention | $\text{num\_head} = 1, \text{hiddendim} = 128$ |

