# OpenReview forum: "MLP-based architecture with variable length input for automatic speech recognition"
_ICLR.cc/2022/Conference — ICLR 2022 Submitted_

### Official Review · Reviewer_uaQ8 · 2021-10-30

**Correctness:** 3
**Technical Novelty And Significance:** 2
**Empirical Novelty And Significance:** 2
**Recommendation:** 6
**Confidence:** 2

**Main Review:**

Please revise the paper in terms of writing. Equation 1 seems to have typo. In section 2, please correct the repetition "the long range dependencies"
It is mistakenly referred to Table 1 instead of Table 2.
- how do we split the dimension in SGU? Randomly?
- you may want to elaborate more on the term "tiny attention"
- have you run the conventional CTC model? Would be good to see that result as baseline too.

**Summary Of The Paper:**

The paper proposes three mlp based structures to deal with sequences of variable size. The experiments are conducted on two datasets and campare to transformer based models some improvements are reported.

**Summary Of The Review:**

While it seems the paper presents technically interesting topics, it requires major revision in terms of writing. It includes many grammatical errors, wrong references and punctuation are not respected. Unfortunately, it makes the paper difficult to follow.

---

> ### Author Response · Authors · 2021-11-17
> **Response to Reviewer uaQ8**
>
> Thank you for your valuable comments.
>
> We are currently revising the manuscript to make the content clearer and to correct grammatical errors. We will upload a new one soon. In addition, we will share the code in a zip file for reproductivity.
>
> > how do we split the dimension in SGU? Randomly?
>
> We split the input in half along the channel dimension, i.e., let X be the DxN input matrix with the first and second dimensions being that of channels and tokens, respectively. Then X_r contains rows 1 to D/2 of X, and X_g rows D/2+1 to D.
>
> > you may want to elaborate more on the term "tiny attention”
>
> We agree that the tiny attention was not sufficiently described and added extra details in Section 4.1.
>
> > have you run the conventional CTC model? Would be good to see that result as baseline too
>
> We are not completely sure which conventional CTC model is referred to. Do you mean the RNN-based CTC model that was first proposed in [a] ? The performance of the transformer encoder CTC model is excellent and it has become the de facto standard, e.g. [b], [c] [d]. We think this is the appropriate baseline for this work.
>
> [a] Alex Graves and Navdeep Jaitly, “Towards end-to-end speech recognition with recurrent  neural networks.  Proc. International Conference on Machine Learning (ICML), pp. 1764–1772. PMLR, 2014.
> http://proceedings.mlr.press/v32/graves14.pdf
>
> [b] Jaesong Lee and Shinji Watanabe, “Intermediate loss regularization for CTC-based speech recognition,” in Proc. ICASSP, 2021, pp. 6224–6228.
> https://arxiv.org/abs/2102.03216
>
> [c ]Ye Bai, Jiangyan Yi, Jianhua Tao, Zhengkun Tian, Zhengqi Wen, and Shuai Zhang, “Listen attentively, and spell once: Whole sentence generation via a non-autoregressive architecture for low-latency speech recognition,” in Proc. Interspeech, 2020, pp. 3381–3385. https://arxiv.org/abs/2005.04862
>
> [d] Ethan A Chi, Julian Salazar, and Katrin Kirchhoff, “AlignRefine: Non-autoregressive speech recognition via iterative realignment,” in Proc. NAACL-HLT, 2021, pp. 1920–1927.
> https://aclanthology.org/2021.naacl-main.154/

---

> > ### Comment · Reviewer_uaQ8 · 2021-11-29
> > **much better now**
> >
> > I appreciate the authors effort in making a better version of the paper. I would like to increase my score now.

---

> > > ### Author Response · Authors · 2021-11-29
> > > **Response to Reviewer uaQ8**
> > >
> > > Thank your for your time and effort towards making this paper better than it was originally! We really appreciate it!

---

### Official Review · Reviewer_4cme · 2021-11-02

**Correctness:** 3
**Technical Novelty And Significance:** 2
**Empirical Novelty And Significance:** 2
**Recommendation:** 5
**Confidence:** 3

**Main Review:**

(1)	The novelty of the paper is somewhat weak. C-MLP, which exploits a depth-wise convolution for time-axis feature aggregation, is a commonly used approach in ASR. (Convolution module in Conformer, or Time-Depth Separable Convolution) TS-MLP seems to bring an idea from S^2-MLP. An additional tiny-attention module is also taken from gMLP.

(2)	For the baseline performance, is there any reason for using only a clean-100h subset of LibriSpeech? I’m not sure the performance is in a sufficient range; it would be clear if you could provide some references that also trained the Transformer only on clean-100h.

(3)	Not enough experimental/architectural details are provided. For example, the embedding dimension, the number of heads in Transformer, convolution kernel size, frontend subsampling stride, type of activation function, STFT window size, optimizer, learning rate, etc., are missing.


**Summary Of The Paper:**

This paper proposes three architectural modifications on recently introduced MLP-based neural networks, such as MLP-Mixer or gMLP. These networks could not handle the variable-length input because the token-level mixing was performed by MLP. The proposed three modifications are: C-MLP (w/ depth-wise convolution), TS-MLP (w/ shift operator), and F-MLP (w/ Fourier transform and w/o gating). The authors evaluated their model on two ASR datasets, LibriSpeech-100h and TedLium-2, and achieved better WER compared to Transformer-based models with a similar number of parameters.

**Summary Of The Review:**

Although the problem that this paper tries to solve is an important problem, the novelty of the proposed solutions seems to be weak. Experiments could be more powerful with commonly used benchmarks (LibriSpeech-960h), and details are missing.

---

> ### Author Response · Authors · 2021-11-17
> **Response to Reviewer 4cme**
>
> Thank you for your valuable comments.
>
> We are currently revising the manuscript to make the content clearer and to correct grammatical errors. We will upload a new one soon. In addition, we will share the code in a zip file for reproductivity.
>
> > (1) The novelty of the paper is somewhat weak. C-MLP, which exploits a depth-wise convolution for time-axis feature aggregation, is a commonly used approach in ASR. (Convolution module in Conformer, or Time-Depth Separable Convolution) TS-MLP seems to bring an idea from S^2-MLP. An additional tiny-attention module is also taken from gMLP.
>
> We fully recognize the ideas borrowed in our paper, (a) the MLP architectures from MLP-Mixer and gMLP (b) time-tested convolution to make the network length agnostic and shift invariant (c) time-shift from S^2-MLP. The novelty of our approach comes from combining these parts into original networks following the spirit of the all-MLP approach, but compatible with variable length sequences. As attested by the number of combinations tested, this was not a trivial task. As a result, this is the first work applying all-MLP networks to ASR. Moreover, the results obtained are not only competitive with, but surpass the performance of the celebrated self-attention architecture with fewer parameters. Our experiments are extensive and include runtime and memory costs analysis. As a consequence, we contest that our work is not sufficiently novel.
>
> > (2) For the baseline performance, is there any reason for using only a clean-100h subset of LibriSpeech? I’m not sure the performance is in a sufficient range; it would be clear if you could provide some references that also trained the Transformer only on clean-100h.
>
> We agree that providing results on the full LibriSpeech-960h would make the paper stronger. We are currently training the networks on the dataset but the training time is longer than the two weeks rebuttal period and we won’t be able to include the results in the revised manuscript. We would include these results in the final version of the manuscript. We point out that many published works, e.g. [a][b][c], include solely experiments on clean-100h.
>
> [a] Higuchi, Yosuke, et al. "A Comparative Study on Non-Autoregressive Modelings for Speech-to-Text Generation." Proc. ASRU, 2021 (to appear).
> https://arxiv.org/abs/2110.05249
>
> [b] D. S. Park et al., “Improved Noisy Student Training for Automatic Speech Recognition,” Proc. INTERSPEECH, 2020. https://arxiv.org/abs/2005.09629
>
> [c] Zhao, Yingzhu, et al. "Cross Attention with Monotonic Alignment for Speech Transformer." Proc. INTERSPEECH. 2020. http://www.interspeech2020.org/uploadfile/pdf/Thu-3-10-8.pdf
>
>
> > (3) Not enough experimental/architectural details are provided. For example, the embedding dimension, the number of heads in Transformer, convolution kernel size, frontend subsampling stride, type of activation function, STFT window size, optimizer, learning rate, etc., are missing.
>
> We added the experimental/architectural details. Our paper is under revision and we will upload a new one soon. In addition, we will share the code in a zip file for reproductivity.

---

### Official Review · Reviewer_h5Ds · 2021-11-02

**Correctness:** 2
**Technical Novelty And Significance:** 2
**Empirical Novelty And Significance:** 2
**Recommendation:** 3
**Confidence:** 4

**Main Review:**

The paper contains many typos and ungrammatical sentences which make it difficult to read. Therefore, even if the experiments seem rigorous,  the lack of rigor in the writing casts a doubt on the results. The reference to previous or related work is scarce and the results of a standard ASR methods that can deal with variable input length should be reported for comparison.
The experiments are not easily reproductible due to the lack of details and errors in the description of proposed models.

The quality of the paper is not sufficient for publication in ICLR.


**Summary Of The Paper:**

This paper deals with the problem of having variable length inputs in deep neural network architecture using only MLPs. Recently, there has been interest in these architectures because they could reduce the number of parameters in deep NN models. However, these architectures have so far only been applied to images with a fixed size input. In order to apply them to sequences such as speech recognition or NLP, these models must be able to deal with variable input size. This paper presents 3 different solutions to allow MLB based deep NN to cope with variable input. The proposed solutions are based on either a modification of the gating mechanism or replace the gating mechanism with FFT. The proposed models are tested for a speech recognition task on 2 standard databases. Compared to transformer-based models, the proposed models obtain comparable performance with less parameters, and better performance with similar number of parameters.

**Summary Of The Review:**

The paper contains many typos and ungrammatical sentences which make it difficult to read. To list a few:
* Page 2: "The long range dependencies The latter allows to model long-range dependencies just like self-attention would.
* Page 3:  "MLP-Mixer simply applies the MLP across the token dimension gMLP has a module called Spatial Gating Unit (SGU)"
* Page 6: "For Librispeech, We tokenize"


* The related work section is very short, focuses only on models for images and gives no reference to ASR.

* Section 2
    * Equation 1, `Linear_{channel}` is referenced in the following sentence as `Linear_{token}`
    * Equations in 2.1 are wrong or wrongly written. Some variable are not explained, and the dimensions don't add up when you consider all the projections.
    * SGU is presented in section 2.2 but has been used in equations of section 2.1
* Section 3
  *  C-MLP', which will be the best model, is not clearly descibed (only equation 8)
  *  "This operation to obtain filter HCGU is independent of the channel dimension." : this is unclear
* Section 4
  * In the text, reported WER seem to have been copied from one to another, and don't match the table.
  * `C-MLP achieves competitive performance with Transformer-based model with only 57.1 % of its parameters`, but after computing the value, it seems to be `57.4 %`
  * Should be `Table 2` not `Table 1` in section 4.4
  * C-MLP' is the best model based on table 1 but is not presented in  table 2
  * In table 5, RTF is not defined
  * In the claims (abstract), it is said that the proposed architecture reduces WER by 1.9/3.4 % but this result seems to be mainly due to the tiny attention module, which is not clearly stated.

---

> ### Author Response · Authors · 2021-11-17
> **Response to Reviewer h5Ds**
>
> Thank you for your valuable comments.
>
> We are currently revising the manuscript to make the content clearer and to correct grammatical errors. We will upload a new one soon. In addition, we will share the code in a zip file for reproductivity.
>
> > The paper contains many typos and ungrammatical sentences which make it difficult to read. Therefore, even if the experiments seem rigorous, the lack of rigor in the writing casts a doubt on the results.
>
> We apologize for these errors and fixed them in the revised manuscript.
>
> > The reference to previous or related work is scarce and the results of a standard ASR methods that can deal with variable input length should be reported for comparison.
>
> We added a detailed section of related works for ASR.  The Transformer baseline compare against is a strong baseline for non-autoregressive ASR used in many prior works [a][b][c][d]. It can deal with variable length input thanks to the self-attention mechanism.
>
> [a] Higuchi, Yosuke, et al. "A Comparative Study on Non-Autoregressive Modelings for Speech-to-Text Generation." Proc. ASRU, 2021 (to appear).
> https://arxiv.org/abs/2110.05249
>
> [b] Jaesong Lee and Shinji Watanabe, “Intermediate loss regularization for CTC-based speech recognition,” in Proc. ICASSP, 2021, pp. 6224–6228.
> https://arxiv.org/abs/2102.03216
>
> [c] Ye Bai, Jiangyan Yi, Jianhua Tao, Zhengkun Tian, Zhengqi Wen, and Shuai Zhang, “Listen attentively, and spell once: Whole sentence generation via a non-autoregressive architecture for low-latency speech recognition,” in Proc. Interspeech, 2020, pp. 3381–3385. https://arxiv.org/abs/2005.04862
>
> [d] Ethan A Chi, Julian Salazar, and Katrin Kirchhoff, “AlignRefine: Non-autoregressive speech recognition via iterative realignment,” in Proc. NAACL-HLT, 2021, pp. 1920–1927.
> https://aclanthology.org/2021.naacl-main.154/
>
>
> > The experiments are not easily reproductible due to the lack of details and errors in the description of proposed models.
>
> We added the experimental/architectural details. In addition, we will share the code in a zip file for reproductivity.
>
> > Section 2
> Equation 1, Linear_{channel} is referenced in the following sentence as Linear_{token}.
> Equations in 2.1 are wrong or wrongly written. Some variable are not explained, and the dimensions don't add up when you consider all the projections.
> SGU is presented in section 2.2 but has been used in equations of section 2.1.
>
> We agree that Section 2 can benefit from significant improvement. We have thoroughly revised it and we will update the manuscript soon.
>
> > Section 3
> C-MLP', which will be the best model, is not clearly descibed (only equation 8)
>
> We added a figure and explanation for 'C-MLP'.
>
> > "This operation to obtain filter HCGU is independent of the channel dimension." : this is unclear.
>
> We were trying to explain that the CGU operation does not mix information between channels. We rewrote this part to make this clear.
>
> > Section 4
> In the text, reported WER seems to have been copied from one to another, and don't match the table.
>
> We apologize for these errors. This is fixed in the revised manuscript.
>
> > C-MLP achieves competitive performance with Transformer-based model with only 57.1% of its parameters, but after computing the value, it seems to be 57.4%.
>
> You are correct! We apologize for the error and fixed this in the revised manuscript.
>
> > Should be Table 2 not Table 1 in section 4.4.
>
> Thank you! We fixed it.
>
> > In table 5, RTF is not defined.
>
>  In Figure 5,  the vertical axis was incorrectly labeled as RTF.  We changed the axis labels in figure 5 to sequence length [s] and inference speed [s].
>
> > In the claims (abstract), it is said that the proposed architecture reduces WER by 1.9/3.4% but this result seems to be mainly due to the tiny attention module, which is not clearly stated.
>
> We rewrote the abstract and now provide the results with and without the tiny self-attention module separately.

---

### Official Review · Reviewer_vrtS · 2021-11-04

**Correctness:** 4
**Technical Novelty And Significance:** 3
**Empirical Novelty And Significance:** 3
**Recommendation:** 6
**Confidence:** 3

**Main Review:**

Strengths:
 - The approach is novel and significant. The MLP approach for image classification is very promising and it's a important step to make it work on sequential data and speech-related tasks.
- The evaluation is thorough, clearly showing the capability of the proposed approaches. The model scaling analysis is a very nice addition and clear showcases the potiental of the MLP architectures.

Weaknesses:
- The clarity of the paper is quite bad. Section 2 especially is hard to read and hard to understand (see detailed comments below). Section 3 is also not very clear. They also are a lot ot typos. The author should improve the writing, as these sections are critical to understand the paper. The experimental setup section is also missing important details about data split, training details and model hyper-parameters.
- I am not convinced about the use of the terms "channel" and "token" for the two dimensions of the input. These keywords are use mainly in CV for the first one and in NLP for the second one. For speech, these two dimensions already have a name: the temporal dimension and the spectral (or cepstral depending on the features) dimension. Moreover, "token" can be misleading as it usually refers to words, but in this case the input is an audio signal segmented by duration, so there are no word boundaries involved. I thus encourage the authors to revisit the naming scheme.

Detailed comments:
- Section 2:
    - "The long range dependencies The latter allows to model ..." -> "The latter allows to model ..."
    - Figure 2: "Linear_toekn" -> "Linear_token"
    - Equation (1) is using "Linear_channel()" but the text mentioned "Linear_token()", which one is correct?
    - SGU is used in Section 2.1 but only explained in Section 2.2. It's confusing.
- Section 4.4:
    - Table 1 should be Table 2
- The term "sandwiched" is used several time, it's not wrong but there are better ways to write that, like "in between" for instance.
- I am a bit confused about the FNet and GFNet baselines in Table 1: according to Section 2 and 3, these two approaches are not suitable for variable length input (which is the key motivation of the paper), so how did the author make them work for ASR? please clarify.

**Summary Of The Paper:**

This paper presents an approach for Automatic Speech Recognition based on MLP architectures. Those architectures were recently proposed for image classification and yielded promising results. In this paper, three new MLP-based architectures able to handle variable length sequences are proposed. The approaches are then evaluated on the Librispeech and Tedlium corpora and are compared to baselines from the literature. The proposed approaches are shown to yield better peformance than the baseline while keeping a low complexity.

**Summary Of The Review:**

The proposed approaches are novel and significant, and clearly evaluated. They are shown to be a promising new approach for ASR. However the clarity and quality of writing of the paper are quite below the standards.  Hence in the current shape i cannot recommend acceptance, it's just below the threshold. I will be happy to increase my score if the writing is improved.

---

> ### Author Response · Authors · 2021-11-17
> **Response to Reviewer vrtS**
>
> Thank you for your valuable comments.
>
> We are currently revising the manuscript to make the content clearer and to correct grammatical errors. We will upload a new one soon. In addition, we will share the code in a zip file for reproductivity.
>
> > The clarity of the paper is quite bad. Section 2 especially is hard to read and hard to understand (see detailed comments below). Section 3 is also not very clear. There also are a lot of typos. The author should improve the writing, as these sections are critical to understand the paper. The experimental setup section is also missing important details about data split, training details and model hyper-parameters.
>
> We have rewritten the whole section 2 and section 3 to make it clearer, including the errors you pointed out. We added the experimental/architectural details in section 4.
>
> > I am not convinced about the use of the terms "channel" and "token" for the two dimensions of the input. These keywords are use mainly in CV for the first one and in NLP for the second one. For speech, these two dimensions already have a name: the temporal dimension and the spectral (or cepstral depending on the features) dimension. Moreover, "token" can be misleading as it usually refers to words, but in this case the input is an audio signal segmented by duration, so there are no word boundaries involved. I thus encourage the authors to revisit the naming scheme.
>
> The names “channel” and “token” are those used in both the MLP-Mixer and gMLP papers which are the main prior work on which we rely in our work. We believe keeping these terms is important for consistency with these works. We have added a short section in the introduction to clarify the relationship of the original temporal/spectral dimensions of the spectrogram to tokens and channels.
>
> > The term "sandwiched" is used several time, it's not wrong but there are better ways to write that, like "in between" for instance.
>
> As suggested, we changed the text to avoid the use of “sandwiched”.
>
> > I am a bit confused about the FNet and GFNet baselines in Table 1: according to Section 2 and 3, these two approaches are not suitable for variable length input (which is the key motivation of the paper), so how did the author make them work for ASR? Please clarify.
>
> Like MLP-Mixer and gMLP, GFNet is designed for inputs of fixed size. In the experiments with GFNet, we zero-pad all the inputs to match the size of the longest sequence in the dataset. This approach is only possible thanks to the moderate number of parameters of GFNet and could not be applied to MLP-Mixer and gMLP (using a reasonable number of parameters).
> Unlike these approaches, FNet can be applied to sequences of arbitrary length as it does not have any learnable parameters for the token mixing operation, using the FFT instead (it was originally proposed for NLP tasks). However, its performance for ASR is not as competitive (as seen in Table 1) as that of the MLP-like architectures we propose, which do have a token mixing module with learnable parameters.
> We have summarized these arguments at the end of Section 2.

---

> > ### Comment · Reviewer_vrtS · 2021-11-30
> > **Response to authors' rebuttal**
> >
> > The authors did a good job improving the clarity of the paper, it's is much clearer now, specially Section 2 and 3. Hence i will increase my score to 6 - just above the thresold.
> >
> > The main reason for keeping my score at 6 is the limited novelty of the proposed approach, as the other reviewers also pointed out, because the paper is mainly an application of a known technique to a new use-case.
> >
> > About the naming choice of keeping "channel" and "token" for clarity, it does actually depend on the target audience of the paper. If the target audience is the image community which knows the MLP-Mixer paper, then I agree. But if the target audience is the speech community, then i think it would be better to use terms the community is familiar with instead of using terms from papers outside of it.

---

> > > ### Author Response · Authors · 2021-12-02
> > > **Response to Reviewer vrtS**
> > >
> > > Thank you for your comment and the score update!

---

### Comment · Area_Chair_xBF4 · 2021-11-15
**Please address reviewers' comments**

Dear Authors,

Please address the reviewers' comments. Thanks!

---

> ### Author Response · Authors · 2021-11-16
> **Response to Area Chair xBF4**
>
> We will post our replies shortly. Thank you for your patience.

---

### Author Response · Authors · 2021-11-19
**We uploaded a revised version.**

We thank reviewers for their valuable comments and suggestions.

We have revised our paper and uploaded a new version.
In addition, we shared the code in a zip file for reproductivity.

As pointed out by many reviewers, there were some major organizational problems with the first draft. In particular, section 2 and section 3 were substantially restructured: the discussion of the MLP-based architecture was divided into two parts: the overall structure of each layer and the description of the token mixing module, which is important in the MLP-based model.

The description for reproduction was also inadequate.
We added the experimental and architectural settings to section 4.
We also summarize the experimental conditions and the preliminary experiments used to determine the proposed architectures in the Appendix, so that anyone can conduct additional experiments and follow-up. Furthermore, full details are available in the code.

The reviewers are also concerned about the novelty of this work.
To emphasize the novelty of this paper, we have also added a description of the issues to applying conventional methods to variable sequence length input data in section 2.3. To the best of our knowledge, the proposed method is the first paper that attempts to apply an MLP-based architecture to ASR. By thoroughly investigating how channel-mixing and token-mixing should be layered, the paper provides new knowledge necessary for the future development of MLP-based models.

Please find below a list of the detailed changes to each section.

abstract
* We describe the results with and without the tiny self-attention module separately.

section1
* We added an explanation of the correspondence between the names "channel" / "token" for images and the term"spectral" / "temporal" for audio data.
* We added a description of related works of ASR.

section2
* We've changed the entire structure of this section to make it clearer.
* We describe two types of architectural configurations of MLP-based architecture.
* We explain the token-mixing module, which is important in the MLP-based model.
* We described the challenges for applying the MLP-based architecture to variable sequence length input data.

section3
* It has been rewritten extensively to make it clearer.
* We first proposed methods to mix the information between tokens and then explained the overall architecture.
* We were pointed out that there is not enough explanation of 'C-MLP', so we added a figure and more explanation.

section4
* We added details of experimental and architectural settings.
* We fixed some errors, such as the results written in the text not matching the table.
* We corrected a reference error in the Table. C-MLP' was not included in Table 2 because its results were inferior to those of C-MLP. In the new version, we included it in Table 2.
* In Figure 5,  the vertical axis was incorrectly written as RTF.  We changed figure 5 to a figure of sequence length [s] and inference speed [s].

---

> ### Author Response · Authors · 2021-11-19
> **We also uploaded experimental codes.**
>
> The experimental codes have been uploaded as supplementary material. Please check them.

---

### Comment · Area_Chair_xBF4 · 2021-11-24
**Please update your ratings if needed based on the authors' responses**

Dear Reviewers,

The authors have made detailed responses to all the reviews. Please take a look and see whether they address your concerns and update the ratings if necessary. Thanks for your help and expertise!

---

### Decision · Program_Chairs · 2022-01-20

**Decision:**

Reject

**Comment:**

This paper adopts the recently developed MLP-based architectures for image classification to Automatic Speech Recognition with 3 different modifications to handle variable length sequences. The three architectural modifications are: C-MLP (w/ depthwise convolution), TS-MLP (w/ shift operator), and F-MLP (w/ Fourier transform and w/o gating). The approaches are then evaluated on the Librispeech-100h and Tedlium-2 corpora and are compared to baselines from the literature. The proposed approaches are shown to yield better performance than Transformer-based models.

As pointed by the reviewers, there are 3 major concerns:
clarity: the initial version of the paper needed more improvement in writing, the authors did improve the writing a lot, which led to increased ratings by the reviewers;
reproduction: many experimental details were missing in the initial version, but the authors added those in the revision and shared the code
novelty: as agreed by all the reviewers that The novelty of the paper is somewhat weak. It is mainly an application of a known technique to a new use-case and the modifications are commonly used in ASR.

The decision is mainly due to the limited novelty.